# Transcriptomics and proteomics reveal two waves of translational repression during the maturation of malaria parasite sporozoites

Scott E. Lindner [1,7,8]*, Kristian E. Swearingen [2,7], Melanie J. Shears [3,6], Michael P. Walker [1], Erin N. Vrana [1], Kevin J. Hart [1], Allen M. Minns [1], Photini Sinnis [3], Robert L. Moritz [2] & Stefan H.I. Kappe [4,5,8]*

*Plasmodium* sporozoites are transmitted from infected mosquitoes to mammals, and must navigate the host skin and vasculature to infect the liver. This journey requires distinct proteomes. Here, we report the dynamic transcriptomes and proteomes of both oocyst sporozoites and salivary gland sporozoites in both rodent-infectious *Plasmodium yoelii* parasites and human-infectious *Plasmodium falciparum* parasites. The data robustly define mRNAs and proteins that are upregulated in oocyst sporozoites (UOS) or upregulated in infectious sporozoites (UIS) within the salivary glands, including many that are essential for sporozoite functions in the vector and host. Moreover, we find that malaria parasites use two overlapping, extensive, and independent programs of translational repression across sporozoite maturation to temporally regulate protein expression. Together with gene-specific validation experiments, these data indicate that two waves of translational repression are implemented and relieved at different times during sporozoite maturation, migration and infection, thus promoting their successful development and vector-to-host transition.

[1] Department of Biochemistry and Molecular Biology, The Huck Center for Malaria Research, Pennsylvania State University, W225 Millennium Science Complex, University Park, PA 16802, USA. [2] Institute for Systems Biology, 401 Terry Avenue N., Seattle, WA 98109, USA. [3] Department of Molecular Microbiology & Immunology, Johns Hopkins Bloomberg School of Public Health, 615 N. Wolfe Street, Baltimore, MD 21205, USA. [4] Center for Global Infectious Disease Research, Seattle Children's Research Institute, 307 Westlake Avenue N. Suite 500, Seattle, WA 98109, USA. [5] Department of Global Health, University of Washington, Seattle, WA, USA. [6] Present address: Department of Laboratory Medicine, University of Washington, 1959 NE Pacific St., Seattle, WA 98195, USA. [7] These authors contributed equally: Scott E. Lindner, Kristian E. Swearingen. [8] These authors jointly supervised this work: Scott E. Lindner, Stefan H. I. Kappe. *email: Scott.Lindner@psu.edu; stefan.kappe@seattlechildrens.org

Malaria remains one of the great global health problems today, taking a large toll on people in the tropics and subtropics. This disease, caused by *Plasmodium* parasites, affects over 200 million people annually and kills over 400,000 (WHO World Malaria Report 2018). While a protein-based subunit vaccine (RTS,S) has recently been licensed and is being used for pilot implementation in three Sub-Saharan African countries, its protection has been limited and relatively short-lived in clinical trials[1]. Developing an effective and long-lasting malaria vaccine that prevents infection remains a chief goal that has yet to be achieved. Accomplishing this goal will require greater knowledge of the basic biology and transmission dynamics of the gametocyte stages as well as pre-erythrocytic sporozoite stages and liver stage parasites. Promising whole-parasite vaccine candidates, based on the sporozoite form of the parasite, are on the horizon and might get closer to realizing a protective vaccine[2].

*Plasmodium* parasites are transmitted between mammalian hosts by female *Anopheles* mosquitoes (reviewed in ref. [3]). Following uptake of male and female gametocytes by the mosquito during a blood meal from an infected host, these parasites activate into gametes in the midgut and fertilize by fusion to form a zygote, which then develops into a motile ookinete. This stage burrows through the midgut wall and establishes an oocyst under the basal lamina. Within each oocyst, the parasite undergoes sporogony to produce up to five thousand oocyst sporozoites, which are released and selectively infect the salivary glands[4]. Oocyst sporozoites are weakly infectious if injected directly into a naïve mammalian host[5], but become highly infectious following proteolytic rupture of the oocyst wall and their transit through the mosquito hemocoel. Sporozoites further gain infectivity after invasion of the salivary glands[5,6]. Interestingly, salivary gland sporozoites lose infectivity for the salivary glands, which was demonstrated by experimentally injecting them into the hemocoel of uninfected mosquitoes[7]. Within the glands, sporozoites await transmission as long-lived, poised salivary gland sporozoites, which occurs when the mosquito takes its next blood meal and injects these sporozoites into the skin. Sporozoites then exit the bite site in the skin, locate and enter the vasculature, and passively travel to the liver. Here, they infect hepatocytes and thus initiate the life cycle progression in the mammalian host[8]. Relatively few sporozoites are injected during a mosquito bite[9] and form a liver stage parasites. Thus, this transmission bottleneck has been the focus of intervention efforts using drugs, subunit vaccines, and attenuated whole-parasite vaccines[2].

Fundamental studies of sporozoite biology have informed efforts to inhibit and/or arrest the parasite during pre-erythrocytic infection. For example, in rodent malaria parasites some transcripts are upregulated in infective (salivary gland) sporozoites (UIS genes), a phenomenon that was originally determined for 23 currently annotated genes by subtractive cDNA hybridization[10]. With the advent of microarray-based transcriptomics, a renewed effort to identify both UIS and upregulated in oocyst sporozoites (UOS) genes identified 124 UIS and 47 UOS genes[11]. Interestingly, only 7 of the original 23 UIS genes were confirmed in this expanded study. However, these UIS genes (UIS1, UIS2, UIS3, UIS4, UIS7, UIS16, and UIS28) have proven to encode some of the most important proteins for the transmission and transformation of the sporozoite into a liver stage parasite, as well as for liver stage development. Gene deletions of some UIS genes have been exploited to generate genetically attenuated parasite strains that arrest during liver stage development[12–15].

In addition to transcriptional control, the malaria parasite also imposes translational repression upon specific mRNAs in female gametocytes, and this mechanism has been observed for at least a few mRNAs in salivary gland sporozoites (reviewed in refs. [16,17]). Translational repression allows for the proactive production of mRNAs and restriction of their translation before transmission, and yet enables just-in-time production of these proteins after transmission when they are needed. However, this strategy (high transcription and low/no translation) is energetically costly, and model eukaryotes and human cells have evolved to avoid this gene regulatory combination (the depleted region of Crick Space) except in specific, beneficial situations[18]. In light of this, it is notable that *Plasmodium* parasites have evolved to use translational repression for transmission to the mosquito, which has been clearly observed for mRNAs (e.g., *p28*) in female gametocytes[19]. The mechanisms underlying this have been established most thoroughly in the rodent malaria parasite *Plasmodium berghei*, where DOZI (a DEAD-box RNA helicase orthologous to human DDX6) and CITH (an Lsm14 orthologue) bind, stabilize, and translationally repress specific mRNAs in female gametocytes[19–21]. Recently, the extent of translational repression in *Plasmodium falciparum* female gametocytes was assessed by mass spectrometry-based proteomics and RNA-seq[22]. In this stage, the parasite expresses over 500 transcripts with no evidence for their encoded proteins, with over half of these maternal gene products being uncharacterized. Enriched in this set of translationally repressed mRNAs are those that encode for functions needed post-transmission and include gene families previously shown to be translationally repressed. These data support the model that female gametocytes, despite high-energetic costs, prepare and await transmission by storing and protecting specific mRNAs needed to establish the infection of the mosquito. However, the extent and precise mechanisms of how translational repression is imposed in sporozoites has not been established beyond targeted studies, which suggest that the PUF2 RBP may act upon *cis* elements found within the coding sequence of the *uis4* mRNA[14].

Systems analysis of translational repression in sporozoites requires knowledge of their global transcriptomes and proteomes, but such analyses were greatly restricted due to substantial contamination with material from the mosquito vector and its microbiome in sporozoite samples[23–25] (reviewed in ref. [26,27]). To address this, we have developed a scalable, discontinuous density gradient purification approach for sporozoites that greatly reduces contamination from the mosquito and its microbes[28]. The resulting fully infectious sporozoites have allowed extensive ChIP, transcriptomic (RNA-seq), and proteomic (nano liquid chromatography–mass spectrometry/MS (nanoLC–MS/MS)) analyses of sporozoites[29–35]. These studies demonstrate that "omics" level analyses of sporozoites are now experimentally practical, and thus reopen long standing questions of mechanisms underlying critical sporozoite functions.

Here, we have addressed one of the foremost questions of sporozoite biology: how and when does the sporozoite prepare molecularly for transmission from the mosquito vector to the mammalian host? Using RNA-seq-based transcriptomics and nanoLC–MS/MS-based proteomics, we here characterize both oocyst sporozoites and salivary gland sporozoites of both rodent-infectious (*Plasmodium yoelii*) and human-infectious (*P. falciparum*) species. Together, these data provide a comprehensive assessment of mRNA and protein abundances, provide evidence for extensive post-transcriptional regulation of the most abundant mRNAs, and demonstrate that two distinct and likely orthogonal translational repression programs are active during sporozoite maturation.

## Results

**Dynamic transcriptional regulation in maturing sporozoites**. Important insights into how sporozoites mature and become

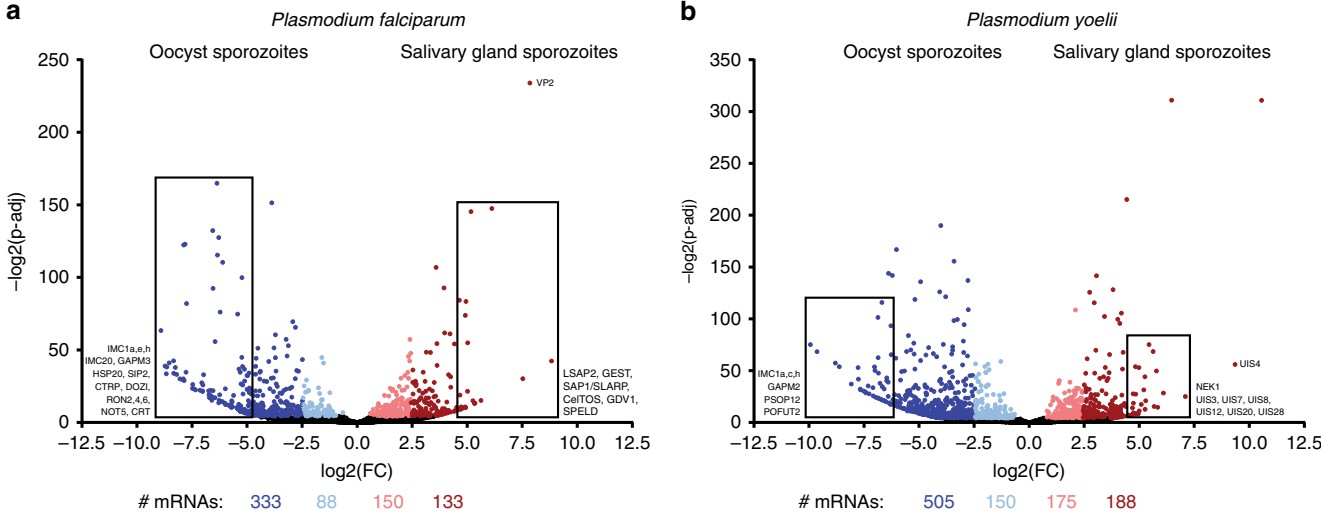

**Fig. 1** Comparative transcriptomics of oocyst and salivary gland sporozoites. RNA from purified **a** *P. falciparum* or **b** *P. yoelii* sporozoites isolated from oocysts or the salivary glands was assessed by RNA-seq, and transcript abundances compared by DEseq2. Transcripts are plotted based upon fold change (between oocyst sporozoites and salivary gland sporozoites) and adjusted *p* value (significance of differential expression), with mRNAs shaded in lighter (±1 to 2.5 log₂ fold change) or darker shades (>2.5 log₂ fold change). Notable, differentially expressed transcripts are labeled

infectious were gained from studies of the upregulated in oocyst sporozoites (UOS) and upregulated in infectious sporozoites (UIS) transcripts in *Plasmodium*. Moreover, a number of these UIS genes turned out to be essential to hepatocyte infection and the early liver stage parasite. However, prior studies were limited by the methods and instrumentation available, thus resulting in an incomplete view of transcriptional regulation in the sporozoite. By leveraging RNA-seq and greatly improved sporozoite purification strategies, we could now achieve a more comprehensive transcriptome and differential expression analyses of sporozoites from rodent-infectious (*P. yoelii*) and human-infectious (*P. falciparum*) species. In addition, as the sporozoite undergoes a transition from being weakly infectious to highly infectious for the mammalian host, which occurs while in transit through the hemocoel from the oocyst to the salivary glands and within the salivary glands[5,6], we assessed both the oocyst sporozoite and salivary gland sporozoite transcriptomes (Fig. 1, Supplementary Table 1, Supplementary Data 1).

First, using *P. yoelii* (17XNL nonlethal strain) rodent-infectious parasites we identified 4195 and 3887 RNAs with detectable and unambiguous sequence reads present in *P. yoelii* oocyst sporozoites and salivary gland sporozoites, respectively. Similarly, with *P. falciparum* (NF54 strain) human-infectious parasites, we identified 3535 and 3575 detectable and unambiguous RNAs in oocyst sporozoite and salivary gland sporozoite stages, respectively. Many well-characterized genes were among the most abundant transcripts in these two stages, including *apical membrane antigen 1 (ama1), circumsporozoite protein (csp), membrane-associated erythrocyte binding-like protein (maebl), perforin-like protein 1 (plp1/spect2), thrombospondin-related anonymous protein (trap), trap-like protein (tlp), UIS 4 (uis4)*, and others (Supplementary Table 1, Supplementary Data 1)[36–41]. Among these, the *maebl* mRNA is known to undergo alternative splicing, which produces a protein with appreciated roles in sporozoite invasion of the salivary glands and a developmentally regulated shift in localization across sporozoite maturation[39,42–44]. Notably, several of the most abundant mRNAs in oocyst and salivary gland sporozoites in both species encode for uncharacterized proteins, some of which (e.g., *py17x_0208200, py17x_0835500*, and *py17x_1354300*) undergo the same extreme

swings in transcript abundance between these stages as does *pyuis4* (>1000-fold). Finally, we found that a recently described sporozoite var gene (SpzPfEMP1) is robustly expressed in not only oocyst sporozoites as previously reported, but also in salivary gland sporozoites and thus may simplify the recently described model of how this interesting var gene is regulated (Supplementary Table 1)[35]. Given the transcript abundance of the novel and uncharacterized genes of these lists, they warrant a prioritized assessment.

Previous definitions of UOS or UIS mRNAs were assigned using lower thresholds of greater than twofold increases in transcript abundance for any detectable transcript between oocyst and salivary gland sporozoites, which in part were dictated by the power of subtractive cDNA hybridization or microarray approaches available at the time[10,11]. With greatly improved approaches, we have made the definitions of UIS and UOS RNAs more stringent by assigning thresholds whereby transcripts must be both in the top decile of abundance, and must be greater than fivefold more abundant in one stage compared to the other. Using these parameters, we have defined 167 UOS mRNAs and 88 UIS mRNAs in *P. yoelii*, and 101 UOS mRNAs and 68 UIS mRNAs in *P. falciparum* (Supplementary Data 2). Few of the UOS transcripts previously defined remain so using these more stringent thresholds, but robustly include the previous top UOS hit: TREP/UOS3[11]. Additional UOS transcripts include those that encode for proteins important for sporozoite functions in the mosquito and the initial infection of a new host, and include those that encode for RNA metabolic processes, protein translation, heat shock proteins (HSP20), the glideosome/inner membrane complex (GAPM3, IMC1m), vesicular trafficking, and transporters. Similarly, a core set of the most abundant UIS transcripts remain defined as such: *uis1, uis2, uis3, uis4, uis7, uis8,* and *uis28*. Strikingly, *pyuis4* transcript abundance increases 1500-fold and reaffirms the use of its promoter for highly enriched expression of transgenes in salivary gland sporozoites[30,45]. An additional 71 *P. yoelii* and 53 *P. falciparum* transcripts that were not in the top decile of RNA abundance, but were in the seventh to ninth decile, increase greater than tenfold in abundance in salivary gland sporozoites vs. oocyst sporozoites. These include *pfccr4*, the *pfdbp10* and *pydbp10* RNA helicases, *pfslarp/pfsap1*

(40-fold), *pynek1* (38-fold), *pyplp3*, *pyplp5*, *pftex150*, *pyuis5* (22-fold), and *pyuis12* (40-fold) (Supplementary Data 3). Beyond the previously defined UIS mRNAs, several other transcripts are notable as their protein products are or may be important to the function of salivary gland sporozoites: FAS-II pathway proteins, fatty acid modifiers, plasma membrane transporters, adhesins/surface proteins (*p113*, *speld*, and *tlp*), traversal-related proteins (*celtos*), heat shock proteins, and ApiAP2 specific transcription factors (e.g., py17x_0523100 and pf3d7_0420300)[46–53]. Together, these transcripts encode for proteins that encompass many essential attributes necessary for sporozoite development, transmission, and infectivity. However, it is striking that while a similar regulatory strategy is employed by sporozoites of both parasite species, there is less overlap in the transcripts that are regulated than might be expected, especially in oocyst sporozoites. These findings underscore the strengths and importance of using comparative and independent approaches with human- and rodent-infectious species to identify the important and conserved molecular components of infection.

**Proteomic comparisons of *Plasmodium* sporozoites.** While transcriptomics can provide an important window into gene expression, the inclusion of proteomics provides a much more comprehensive understanding of the parasite's molecular and cellular functions. To determine the presence and steady-state abundance of proteins found within *Plasmodium* sporozoites, the global proteomes of both *P. yoelii* and *P. falciparum* oocyst sporozoites were determined by nanoLC–MS/MS and were compared to our previously published salivary gland sporozoite global proteomes[31]. This approach (steady-state protein abundance) was used, as it is compatible with currently feasible sporozoite production and purification capabilities, whereas ribosome profiling remains technically unfeasible with sporozoite samples due to the number of highly purified sporozoites that are required. Together, these four datasets now allow a more complete understanding of the oocyst and salivary gland sporozoite stages, and also allow for the definition of UOS Proteins and UIS Proteins for the two distinct malaria parasite species.

Using approximately four million purified sporozoites per biological replicate, protein lysates were separated in a single lane of a gradient sodium dodecyl sulphate (SDS)-polyacrylamide gel, digested with trypsin, and the resulting tryptic peptides were extracted and subjected to nanoLC–MS/MS. Resulting mass spectra were assessed with the trans-proteomic pipeline (TPP) to identify peptides and to infer identities of proteins. In sum, reanalysis of our previously acquired *P. falciparum* data identified 2037 salivary gland sporozoite proteins[31] and we now here also identify 1430 oocyst sporozoite proteins; similarly, from our previously acquired *P. yoelii* data, we identified 1773 salivary gland sporozoite proteins, and here identify 1760 oocyst sporozoite proteins (Fig. 2, Supplementary Table 2, see Supplementary Data 1 for a complete list). As has been shown in previous interspecies comparisons of sporozoite proteomes (i.e., *P. falciparum* and *P. vivax*[32]), we observed a core group of high-abundance, essential proteins that were similarly expressed in both *P. yoelii* and *P. falciparum* sporozoites. These included well-characterized sporozoite proteins (CSP, CelTOS, TRAP, IMC/glideosome proteins, ALBA proteins, and SIAP1) and abundant housekeeping proteins (histones, HSPs, GAPDH, and translation-related proteins). In addition, dynamic changes in the abundance of specific proteins between the oocyst and salivary gland sporozoite stages were also identified. For instance, in both *P. yoelii* and *P. falciparum*, CelTOS, GEST, and SPELD were not detected or were only weakly expressed in oocyst sporozoites, but were among the most abundant proteins in salivary gland

sporozoites. This coincides with the maturation of sporozoite invasion organelles during sporozoite transition (Supplementary Table 2, Supplementary Data 1, Supplementary Fig. 1). Moreover, the presence/absence of cellular regulators such as specific ApiAP2s, histone modifiers, RBPs, and other proteins (Supplementary Data 1) agree with previous reports describing how these types of regulation may be used by sporozoites[35,46,54].

With proteomic data from both oocyst sporozoites and salivary gland sporozoites, we have expanded the UIS and UOS designations to proteins that are differentially abundant in one stage or the other. These designated UIS and UOS proteins in *P. yoelii* and *P. falciparum* were identified using the same stringent threshold as was applied to mRNA abundances (greater than sixfold more abundant in one stage compared to the other). Moreover, this was applied only to the top half of detected proteins of oocyst sporozoites (for UOS proteins) or salivary gland sporozoites (for UIS proteins), as differences in protein abundances quantified by spectral counting methods are most robust among higher-abundance proteins[55]. Based upon this analysis, we identified 30 UOS proteins and 114 UIS proteins in *P. falciparum*, and 65 UOS and 65 UIS proteins in *P. yoelii*. UOS proteins detected in both species include UOS3/TREP, PCRMP2, and PCRMP4 (Fig. 3, Supplementary Data 4), which all have clearly been shown to be expressed in and are important to oocyst sporozoites[11,56,57]. Similarly, UIS proteins in both species include 6-Cys proteins (P38, P36, P52, and B9), CLAMP, GEST, PLP1/SPECT2, PUF2, Sir2A, SPATR, SPELD, TRAMP, and UIS2, which have roles in salivary gland sporozoite infectivity, enabling the sporozoite to navigate the host skin and liver, or traverse and productively infect hepatocytes. As with differential expression of RNA, species-specific differences in protein abundance changes were observed, with notable proteins being ApiAP2-SP (*P. falciparum*), and SPECT1, UIS3, and GAMER (*P. yoelii*). These data sets include many of the best-characterized sporozoite proteins, which are also known to be critical to sporozoite maturation and transmission. However, in both species and in both stages, 28–49% of the proteins now defined as UOS and UIS proteins remain uncharacterized and are likely to be important to sporozoite functions in the mosquito vector and the mammalian host.

**Comparison of UOS/UIS designations within and across species.** When comparisons across species for UOS RNAs and proteins (Fig. 3a) or UIS RNAs and proteins (Fig. 3b) are made, several key features emerge. First, there are very few gene products that receive the same UOS designations across species (e.g., 14 UOS mRNAs and 6 UOS proteins), few that are both UOS mRNAs and proteins (5 in *P. falciparum*, 7 in *P. yoelii*), and only a single instance of a syntenic gene that encodes a UOS mRNA and protein in both species: TREP/UOS3. However, these proteins are known to be important to the oocyst sporozoite. For example, we identified TREP/UOS3 as well as *Plasmodium* Cysteine Repeat Modular Protein 2 and 4 (PCRMP2 and PCRMP4) as UOS proteins in both *P. yoelii* and *P. falciparum*. TREP/UOS3 and PCRMP2 are important for sporozoite targeting to the salivary gland[11,57], whereas PCRMP4 is important for oocyst egress[56]. Similarly, relatively few gene products receive the same UIS designations across species (Fig. 3b), but those that do include several gene products known to be important to salivary gland sporozoites. For instance, 10 cross-species UIS mRNAs and 25 UIS proteins were detected, and include gene products that enable the sporozoite to preserve its infectivity (PUF2), relieve translational repression (UIS2), traverse through host cells (PLP1 and CelTOS) and more. Two of these gene products are both UIS mRNAs and UIS proteins in both species: CelTOS and SPELD

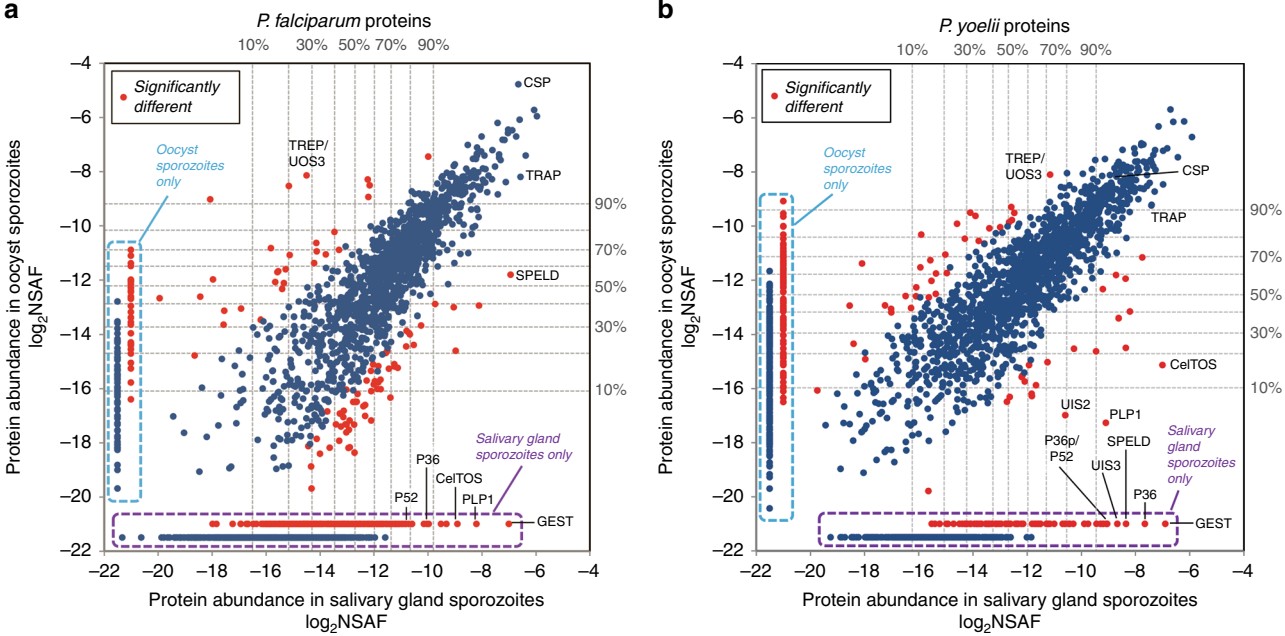

**Fig. 2** Comparative total proteomics of oocyst sporozoites and salivary gland sporozoites. Total protein from purified **a** *P. falciparum* or **b** *P. yoelii* sporozoites isolated from oocysts or salivary glands was separated by a gradient SDS-PAGE, digested with trypsin, extracted, and analyzed by LC–MS. Protein abundance is presented as the $\log_2$ of the normalized spectral abundance factor (NSAF), a label-free quantification approach that enables comparison of relative protein abundance within and between samples. Proteins with statistically significant differences in abundance are noted with red dots, and expression percentile thresholds are provided by dashed lines. Proteins that were only detected in oocyst sporozoites or salivary gland sporozoites are boxed in light blue and purple dashed lines, respectively. Notable proteins are labeled

(Supplementary Data 1). While less is known about SPELD, which is found on the sporozoite surface and is essential for liver stage development, CelTOS has been the focus of substantial study and is a promising target for therapeutic interventions[47,58]. We anticipate that the uncharacterized gene products identified here will play roles in similar processes as those that have already been studied, and deserve prioritization in future work. Together, these data indicate that similar gene regulatory strategies are used by *P. yoelii* and *P. falciparum* sporozoites, but in order to navigate and interact with specific mammalian host environments, they might regulate their most abundant gene products differently. Relaxation of the more stringent thresholds used to define UOS and UIS gene products by inclusion of additional expression deciles and/or requiring a lower fold change yields substantially more overlap in the regulated gene products (Supplementary Data 1).

**Independent translational repression programs in sporozoites.** *Plasmodium* parasites have adopted the use of translational repression in female gametocytes in a manner analogous to the maternal-to-zygotic transition of metazoans, with translation of stored and protected mRNAs occurring post transmission to the mosquito[19]. However, far less is known about whether a similar, energetically unfavorable regulatory strategy is used in sporozoites. Currently, few transcripts have been shown to be translationally repressed in sporozoites through reverse genetic studies. The best-studied example is the *uis4* transcript, which has *cis* control elements located in the coding sequence itself to limit translation of the UIS4 protein prior to transmission[14]. In addition, a translational repressor, PUF2, has been shown to be essential for the preservation of sporozoite infectivity during an extended residence in the salivary glands[30,59–61]. Recently, transcriptomic and proteomic data from *P. vivax* sporozoites has indicated that translational repression occurs in this species as well[62].

In order to identify putatively translationally repressed transcripts in sporozoites, we analyzed our transcriptomic and proteomic data for evidence of highly abundant transcripts for which no protein could be detected. Existing data suggest that translational repression is imperfect, meaning that translationally repressed mRNAs may still produce a detectable amount of protein. Therefore, in these comparisons we used the following highly stringent criteria to define a translationally repressed transcript: (1) transcripts must be in the top decile of mRNA abundance, (2) the corresponding protein must be either undetected or exhibit a disproportionately low abundance (e.g., bottom 50th percentile), and (3) must encode for a protein with detectable tryptic peptides (Supplementary Data 5). Through comparison of the combined RNA-seq and proteomics datasets, we observed that, as expected, transcript and protein abundance correlated well for many essential and conserved gene products, e.g., CSP, TRAP, CelTOS, SPELD, and GEST. However, there was also widespread temporal dysregulation between transcript and protein abundance, including evidence that translational repression is extensively imposed upon many of the most abundant mRNAs of both oocyst sporozoite and salivary gland sporozoite stages of both species (Supplementary Data 6, Supplementary Fig. 2). The extent of translational repression of transcripts in the top decile of abundance is comparable across both species and both sporozoite stages, with each species having transcripts with no evidence (~40–50% of mRNAs), or no or low amounts (~68–80% of mRNAs) of protein detected. Specifically, 115 of 167 UOS mRNAs and 70 of 88 UIS mRNAs are translationally repressed in *P. yoelii*, whereas 62 of 101 UOS mRNAs and 50 of 68 UIS mRNAs are translationally repressed in *P. falciparum*. Complete lists of the top 10% most abundant transcripts that are translationally repressed are provided (Supplementary Data 1 and 6).

Importantly, these datasets also reveal that *Plasmodium* has implemented two discrete and likely orthogonal translational repression programs during sporozoite maturation and transmission. One program imposes translational repression in oocyst

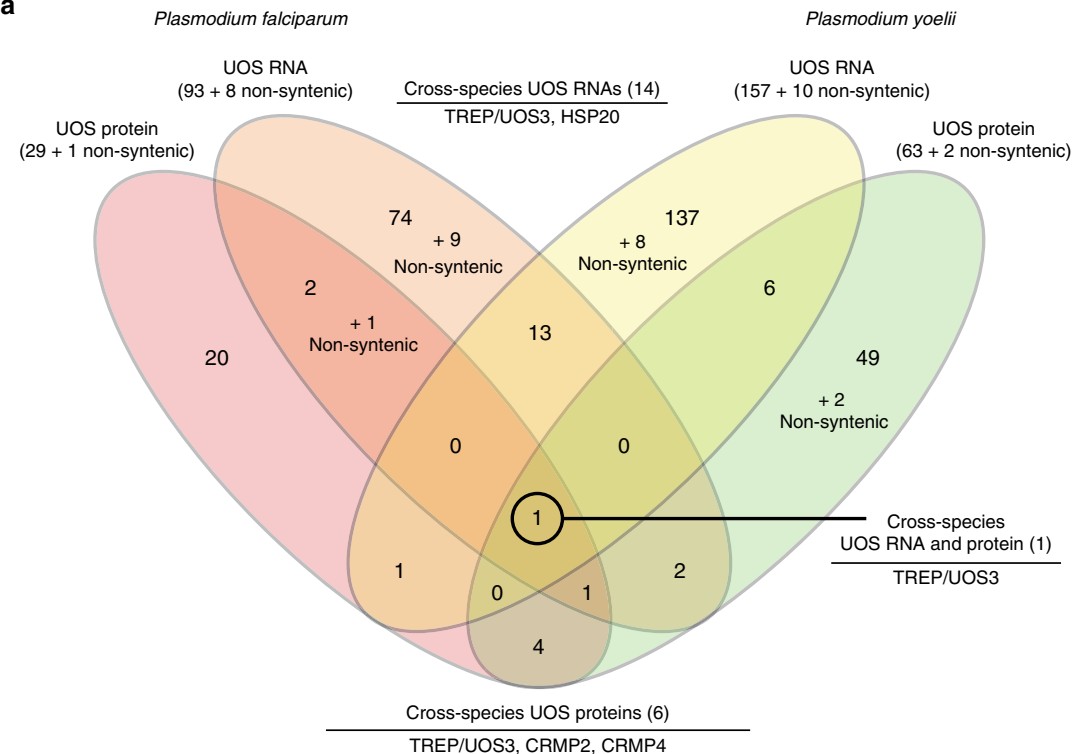

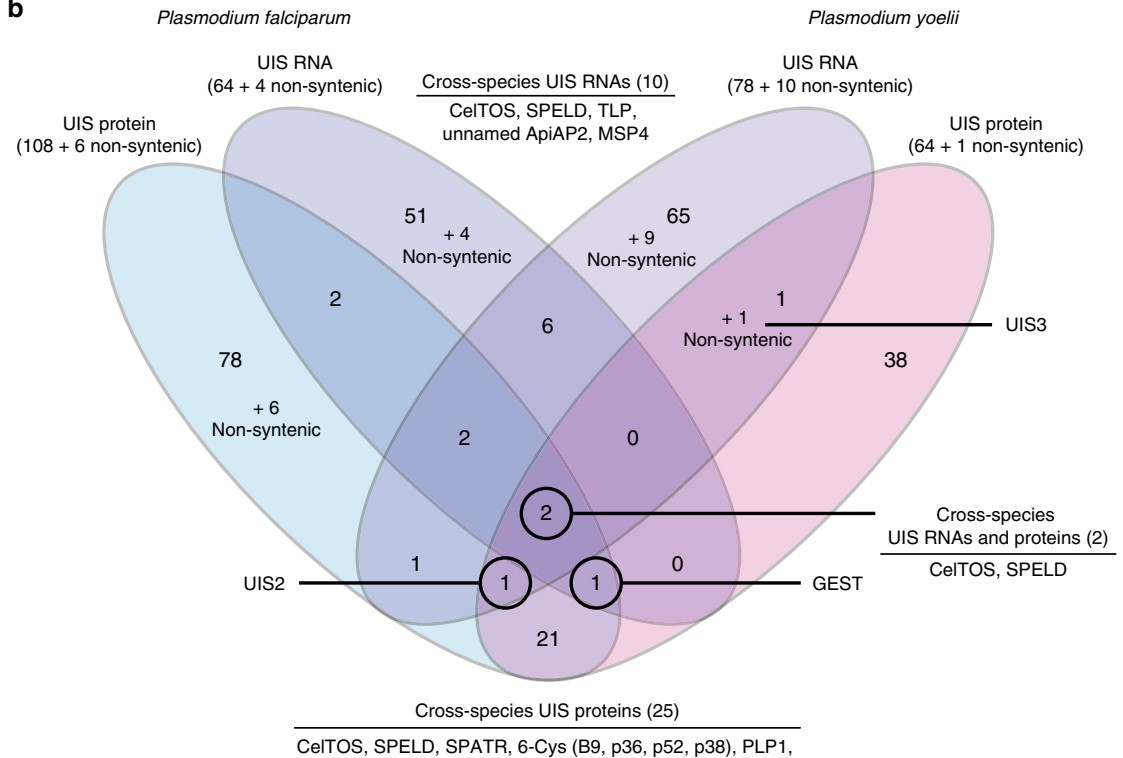

**Fig. 3** Comparisons of UOS and UIS Gene Products. RNAs or proteins in the top decile of abundance that were also at least fivefold (mRNA) or sixfold (protein) more abundant in either oocyst sporozoites or salivary gland sporozoites were denoted as UOS or UIS gene products, respectively. Comparisons across molecule types and species for **a** UOS and **b** UIS identify many gene products that are critical/essential to sporozoite development and/or transmission. Gene products that are similarly regulated across species, transcriptional levels, and/or translational levels are indicated

sporozoites, which is relieved in salivary gland sporozoites to allow for the production of highly abundant proteins (TR-oospz to UIS Proteins program) (Supplementary Table 3). A second program imposes and retains translational repression upon mRNAs throughout sporozoite maturation (pan-sporozoite TR program), which may allow for de-repression in the liver stage parasite as is the case for *pyuis4* (a selected list is provided in Supplementary Table 4; Supplementary Fig. 1). However, formal demonstration of the full scale of a post-transmission release from the pan-sporozoite TR program awaits technical advances to enable total proteomics of early liver stage parasites.

Strikingly, for both programs, well-characterized mRNAs are regulated to allow production of their encoded proteins when they are required for the parasite's activities. For instance, the TR-oospz to UIS Protein program (Supplementary Table 3) controls production of PLP1/SPECT2, CelTOS, and TLP, which are critical or essential for the sporozoite to navigate the host skin, vasculature, and liver[37,52,53,63–65]. Analyses of the complete TR-oospz to UIS Protein dataset reveal significant GO terms noting roles in the apical invasion complex, the parasite cell surface, movement in host environments, and interaction with and entry into host cells (Supplementary Data 6). These data are in full agreement with original studies of PLP1/SPECT2 and CelTOS in *P. berghei*, which used IFA, western blotting, and immuno-EM to show that neither protein is present in oocyst sporozoites, but that both become abundant in salivary gland sporozoites[37,53]. Work on other proteins provide supporting evidence for their expression in and importance to sporozoite functions in the salivary glands and early steps in the infection of the mammalian host[66–68].

Similarly, the Pan-Sporozoite Translational Repression program affects transcripts that encode for proteins that are known to be important/essential for subsequent stages of the parasite, with notable overlapping regulation of ApiAP2-I, MORN1, UIS11, and PAIP1 in both species and with similar timing (Supplementary Table 4). In addition, in *P. yoelii*, several of the historically defined UIS mRNAs (UIS4, UIS8, UIS12 (when including the top two deciles), UIS28), ApiAP2-SP3, ApiAP2-L, and others are regulated by this program. In *P. falciparum*, GAMER, HDAC1, RNA metabolic enzymes, CDPK1, CDPK6, FabZ, ApiAP2-O4, two unnamed ApiAP2s, and other regulator proteins are affected. This indicates that the sporozoite is capable of immediate regulation of these mRNAs before any significant translation can occur, and is consistent with models that position cytosolic granules near the nuclear pore complex to receive exported mRNAs[69]. Taken together, these data indicate that sporozoites have evolved two overlapping and independent translational repression programs to prepare and remain poised for their next required functions in a closely orchestrated manner.

**Validation of translational repression in sporozoites**. To further validate the regulation of select mRNAs by the pan-sporozoite TR program, we have used a gold standard, gene-by-gene assessment of wild-type and transgenic *P. yoelii* salivary gland sporozoites by fluorescence microscopy. For this, we have selected genes that exhibit RNA abundances in either the 99th percentile (UIS4 and PY17X_1354300) or at the 80th percentile (UIS12), but that by mass spectrometry have exceedingly low protein abundances (7, 2, and 1 peptide spectrum matches in salivary gland sporozoites, respectively). Previous characterizations of UIS4 in both *P. yoelii* and *P. berghei* have yielded conflicting data on the presence/abundance of this protein using either fluorescence microscopy or mass spectrometry-based proteomics. To address this, we have generated rabbit polyclonal antisera against recombinant PyUIS4 to monitor protein

abundance in wild-type sporozoites. By IFA, nearly all day 14 salivary gland sporozoites showed no UIS4 protein detectable above background (Fig. 4a). This is consistent with a previous report that showed UIS4 protein levels increase over the residence time of *P. berghei* sporozoites in the salivary gland[70]. Together with our current findings, these data are consistent with a robust but incomplete translational repression of UIS4, which becomes increasingly leaky over time in sporozoites, even in the earliest isolatable salivary gland sporozoites. We hypothesize that this might be attributed to our incomplete understanding of how to minimally perturb sporozoites upon extraction from the mosquito.

We have further investigated whether our classifications of translational repression apply to uncharacterized gene products with mRNAs in the top decile of abundance. To this end, we chose PY17X_1354300, which is one of the most abundant mRNAs in *P. yoelii* salivary gland sporozoites (99.5th percentile) but was among the least abundant proteins detected (Supplementary Data 1). We created PY17X_1354300::GFP transgenic salivary gland sporozoites, and in agreement with the proteomic data, did not detect the presence of PY17X_1354300::GFP protein using anti-GFP antibodies (Fig. 4b). Finally, while we have restricted our definition of translationally repressed transcripts to include only the most abundant mRNAs, it is also likely that less abundant mRNAs are similarly regulated as well. To address this, we assessed PyUIS12 protein expression in salivary gland sporozoites, as it has high mRNA expression (80th percentile) but was barely detected by mass spectrometry (a single peptide spectrum match (PSM) in salivary gland sporozoites). Using live fluorescence microscopy with PyWT-GFP or PyUIS12::GFP sporozoites, we clearly observed GFP expression in control *P. yoelii* WT-GFP sporozoites, but did not detect UIS12::GFP protein when transcribed from its native locus (Fig. 4c). In agreement with translational repression being relieved post-transmission, IFA micrographs clearly show UIS12::GFP expression in the cytosol of 24-h old liver stage parasites (Fig. 4d). Taken together, these data indicate that *Plasmodium* parasites can impose translational repression upon sporozoite transcripts, and can do so beyond what our conservative definition applied to only the top decile of RNA abundance encompasses. However, it is notable that because the consistency and completeness of this regulation varies across individual sporozoites, both global (such as those applied here) and single-cell approaches are informative and required to understand this regulatory process.

## Discussion

*Plasmodium* sporozoites are an intriguing model of parasite infection biology with distinct infectivity profiles in the mosquito vector site of development (oocysts) and site of sequestration for transmission to the mammalian host (salivary glands). Here, we report a comprehensive and comparative assessment of the transcriptomes and proteomes of both *P. yoelii* and *P. falciparum* sporozoites. We have captured these gene expression and protein profiles for immature sporozoites from the mosquito midgut (oocyst sporozoites) and mature, infectious sporozoites from the mosquito salivary glands. From these extensive data sets, several important features of transcriptome and proteome regulation can be deciphered that are likely controlling the distinct sporozoite phenotypes in the mosquito vector and mammalian host.

First, these datasets provide a robust classification of transcript regulation across sporozoite maturation at both the mRNA and protein levels. Previous work identified mRNAs UOS or salivary gland sporozoites, but relied upon less comprehensive instrumentation and low stringency thresholds. The use of current RNA-seq methodologies and improved genome annotation

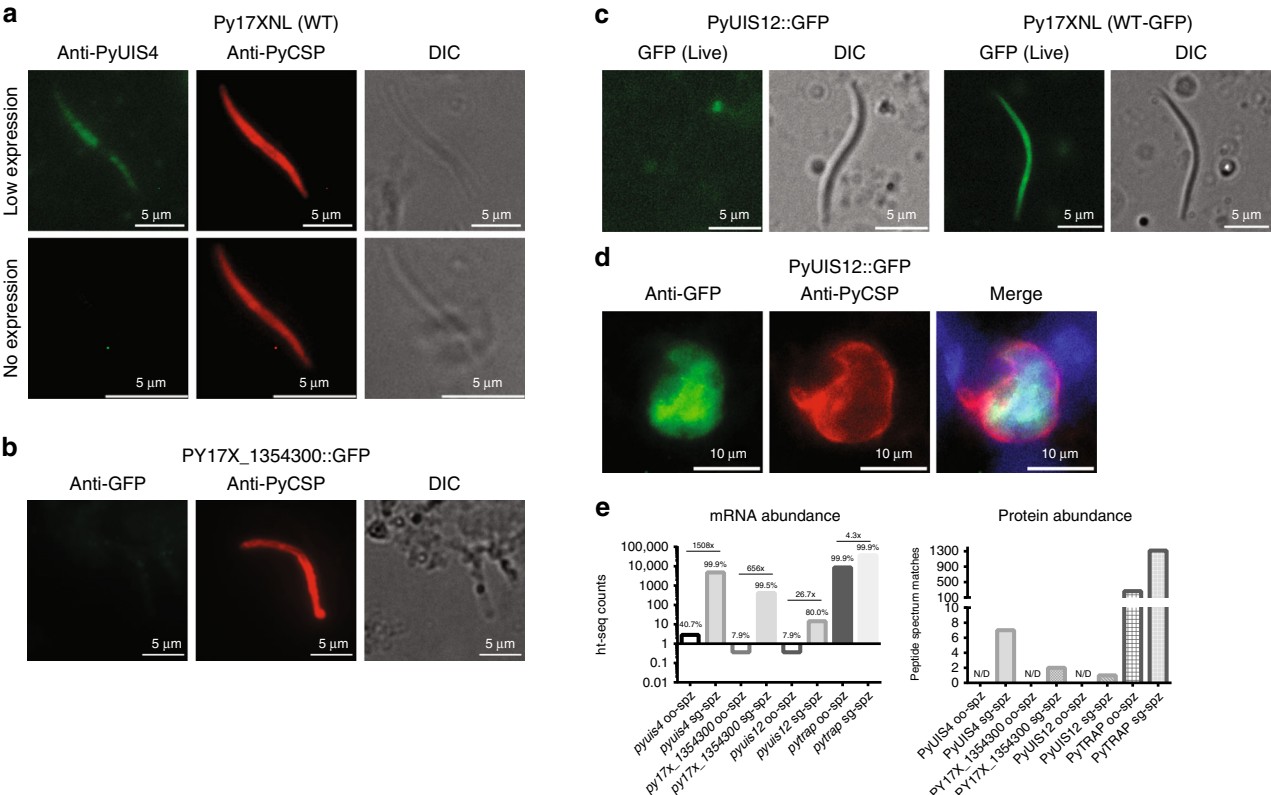

**Fig. 4** Validation of translationally repressed transcripts in *P. yoelii*. The presence and relative abundance of proteins encoded by translationally repressed mRNAs was assessed by fluorescence microscopy. **a** PyUIS4 expression is largely absent in the population of wild-type sporozoites taken shortly after salivary gland invasion. Very few sporozoites have detectable PyUIS4 protein, which is weak and diffuse. **b** An uncharacterized protein, PY17X_1354300, was modified to encode a C-terminal GFP tag, and protein expression was monitored by IFA. In agreement with the total proteomic data, no protein was detectable. **c** Transgenic parasites expressing unfused GFP (WT-GFP control) or GFP fused to UIS12 were assessed by live fluorescence for evidence of protein expression. No PyUIS12::GFP could be detected above background in sporozoites. **d** However, robust expression of PyUIS12::GFP is detected by IFA post-transmission in 24 h liver stage parasites. Scale bars are 5 microns (panels **a**–**c**) or 10 microns (panel **d**). **e** The abundance of mRNA (left) and protein (right) for PyUIS4, PY17X_1354300, and PyUIS12 are provided in comparison to PyTRAP. The percentile of mRNA abundances (based upon ht-seq counts) compared to all other transcripts is provided, with the fold change between oocyst sporozoites and salivary gland sporozoites noted above paired bars. The number of total, unambiguous peptide spectrum matches is provided. N/D: not detected

employed here provides a far more extensive and robust classification of UOS and UIS transcripts, and now does so for both rodent-infectious (*P. yoelii*) and human-infectious (*P. falciparum*) sporozoites (Fig. 1, Supplementary Table 1, Supplementary Datas 1–3). Moreover, we have also assessed these parasites for large-scale changes in protein abundance through mass spectrometry-based proteomics (Fig. 2, Supplementary Table 2, Supplementary Data 1 and 4). Together, these data strongly align with the expression levels and timing reported for individually studied mRNAs and proteins, and will provide the foundation for a systems analysis of the regulatory networks that govern sporozoite infection biology.

Second, we uncovered evidence that extensive translational repression occurs in both *P. falciparum* and *P. yoelii* oocyst sporozoites and salivary gland sporozoites. In analyzing our data, we first applied rigorous thresholds to interrogate the most abundant transcripts and proteins with the goal of identifying putative targets with the highest possible confidence. We deemed this prudent, as detection of mRNAs by RNA-seq (which includes sample amplification approaches) is more sensitive than detection of proteins by mass spectrometry (which cannot benefit from sample amplification). Among the top decile of mRNAs by abundance, the encoded proteins for nearly half were not detected at all by mass spectrometry, and the encoded proteins for another

quarter were detected at a disproportionately low abundance (Supplementary Data 6). It is notable that relaxation of these thresholds reveals that translational repression also occurs with less abundant mRNAs, which we also observed through microscopy in the validation of PyUIS12 expression (Fig. 4).

Intriguingly, we find that two translational repression programs appear to be functioning in sporozoites, with some transcripts being translationally repressed in oocyst sporozoites but highly translated in salivary gland sporozoites (TR-oospz to UIS Protein program) while others remain translationally repressed throughout sporozoite maturation (pan-sporozoite TR program) (Fig. 5). In the case of those proteins that have been characterized for their roles in sporozoite maturation and functions in the mosquito and host, clear patterns arise. The TR-oospz to UIS Protein program would provide for the rapid production of proteins in salivary gland sporozoites, and would be well-suited for proteins that are needed immediately after transmission for host cell traversal in the skin, vasculature and liver, and/or for productive infection of hepatocytes. In agreement with this, we find several proteins with known roles in cell traversal (PLP1/SPECT2, CelTOS, GAMER, TLP; Supplementary Table 3). The second Pan-sporozoite TR program, particularly including those UIS transcripts that are expressed only in salivary gland sporozoites but that are translationally repressed, would regulate the

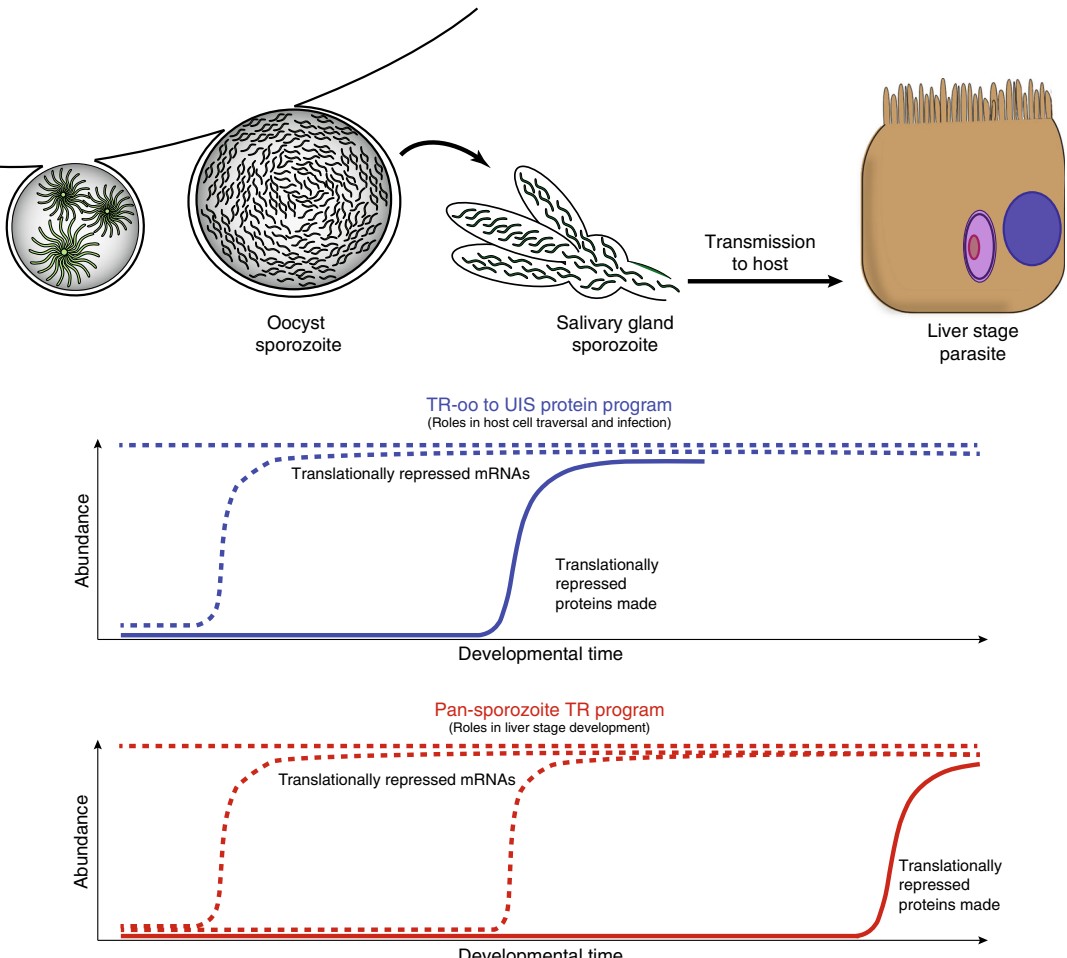

**Fig. 5** A model for two independent translational repression programs in *Plasmodium* sporozoites. One regulatory program acts upon mRNAs that are highly abundant in oocyst sporozoites to impose translational repression, which is relieved in salivary gland sporozoites and enables protein translation to occur. Transcripts encoding for proteins important to host cell traversal and initial infection are regulated in this manner. A second regulatory program translationally represses mRNAs throughout sporozoite maturation, which is hypothesized to be relieved following transmission and infection of a hepatocyte to promote liver stage development

establishment of a new intrahepatocytic liver stage of infection by allowing for the rapid translation of these mRNAs after hepatocyte invasion. In the absence of robust global proteomic analysis of the early liver stage parasite, which will be exceedingly difficult to achieve, this dataset constitutes the best possible platform from which to assess the liver stage proteome in a candidate-based approach.

Together, our findings indicate that a multitiered, temporal translational repression mechanism is at work in *Plasmodium* sporozoites. This regulatory system aligns with the several windows of functionality that are required for the sporozoites' journey as they egress from the relatively benign environment of oocysts on the mosquito midgut, migrate through the hemocoel, invade the salivary glands, remain there poised for transmission and when transmitted, migrate in mammalian tissue, avoid the dangers of the host immune response, traverse cells, and ultimately infect hepatocytes. As the posttranscriptional control of specific mRNAs is energetically unfavorable as compared to de novo transcription, we hypothesize that reducing the time between receipt of an external/environmental stimulus and the availability of a protein is critical to the parasite. The use of a TR-oospz to UIS Protein program is straightforward, as it would require a short duration of translational repression until invasion

of the salivary glands occurs. However, it is less clear why an energetically unfavorable pan-sporozoite program would be activated in oocyst sporozoites, instead of simply transcribing these mRNAs in salivary gland sporozoites. One scenario that could explain the use of both programs is one where the sporozoite invades the salivary gland and then is immediately transmitted, as this would allow immediate responses to both events. While population-level approaches (like those used here) are practical and informative, single-cell approaches should be coupled with them to uncover meaningful differences in the variance of gene expression across individual sporozoites. Enabling technology for single-cell RNA sequencing is currently available, and single-cell proteomics is on the horizon.

Finally, new questions emerge from these data. For instance, what are the *trans* factors and *cis* elements responsible for these two likely orthogonal translational repression systems? While several RNA-binding proteins (RBPs) have been implicated in the preparation of salivary gland sporozoites for transmission, specific RBPs have not been associated with specific transcripts in the sporozoite. Moreover, as the TR-oospz to UIS Protein program initiates in the oocyst sporozoite, experiments must also be pursued in this stage as well. In addition, at what point are mRNAs governed by the pan-sporozoite TR program released for

translation? The prevailing model based upon a few gene-specific examples suggests that it should be relieved after hepatocyte infection in early liver stage. Also, could another program be active in *Plasmodium* species that produce latent liver stage forms, called hypnozoite stages? As the commitment to active or latent liver stage forms might already occur in salivary gland sporozoites, having a translational repression program available in sporozoites could allow for this, and could also contribute to determining the frequency of latent liver stage parasites. Lastly, perhaps the most appealing questions of all revolve around the uncharacterized and under-characterized gene products identified here. These may provide new clues to unappreciated parasite functions, or produce proteins so very different from host proteins that they can be therapeutically targeted.

## Methods

**Plasmodium sporozoite production and purification.** Wild-type *P. yoelii* (17XNL strain) sporozoites were produced in a temperature (24C), humidity (70%), and light (12 h cycles) controlled incubator[30]. Briefly, 6-to-8-week old Swiss Webster mice were infected by intraperitoneal (IP) injection of cryopreserved infected blood and were monitored until the peak day of male gametocyte exflagellation. Mice were then anesthetized with an IP injection of ketamine/xylazine and exposed to 150–200 *Anopheles stephensi* mosquitoes for 15 min with periodic movements on the cage to promote consistency in the transmission of parasites to the mosquito population. Oocyst sporozoites were collected by microdissection and grinding of mosquito midguts on day 10 post-blood meal, whereas salivary gland sporozoites were similarly collected from salivary glands on day 14 post-blood meal.

All animal care adheres to the Association for Assessment and Accreditation of Laboratory Animal Care (AAALAC) guidelines, and all experiments conformed to approved IACUC protocols at the Center for Infectious Disease Research (formerly Seattle Biomedical Research Institute, Protocol ID #SK-02 to Stefan Kappe) or at Pennsylvania State University (Protocol ID #42678 to Scott Lindner). To this end, all work with vertebrate animals was conducted in strict accordance with the recommendations in the Guide for Care and Use of Laboratory Animals of the National Institutes of Health with approved Office for Laboratory Animal Welfare (OLAW) assurance.

Wild-type *Plasmodium falciparum* (NF54 strain) sporozoites were produced[33] by Seattle Children's (formerly the Center for Infectious Disease Research, Seattle Biomedical Research Institute) and Johns Hopkins University. *P. yoelii* and *P. falciparum* sporozoites were purified by DEAE sepharose and/or two sequential Accudenz gradients[28,31,71].

**Reverse genetic modification of *P. yoelii* parasites.** *Plasmodium yoelii* (17XNL strain) was genetically modified using conventional, double homologous recombination approaches with the pDEF plasmid vector[30]. Oligonucleotides used for the creation of targeting sequences are listed in Supplementary Data 7. The 3′ end of *py17X_1354300* or *pyuis12* (PY17X_0507300) was modified by the addition of the GFPmut2 coding sequence prior to the stop codon. Transgenic parasites were identified by genotyping PCR, with independent transgenic clones being isolated by limiting dilution cloning. Clonal parasites were transmitted to *A. stephensi* mosquitoes to produce salivary gland sporozoites as described above.

**Live fluorescence and indirect immunofluorescence assays.** Wild-type and transgenic *P. yoelii* sporozoites (PY17X_1354300::GFP, PyUIS12::GFP) were subjected to live fluorescence assays and/or an indirect immunofluorescence assay (IFA)[72] to characterize the extent of translational repression of these candidates. For live fluorescence microscopy of PyUIS12::GFP, freshly produced salivary gland sporozoites placed on glass slides in VectaShield, overlaid with a cover glass slip, and visualized by fluorescent microscopy using a Zeiss Axioscope A1 with 8-bit AxioCam ICc1 camera and Zen imagine software from the manufacturer. Alternatively, fresh salivary gland sporozoites were fixed in 10% v/v formalin for 10 min, and then air dried to a well on a glass slide defined by a hydrophobic pen. Sporozoites were treated for IFA using either rabbit polyclonal anti-PyUIS4 (antigen consisting of AA80-224, diluted 1:1000), produced by Pocono Rabbit Farm and Laboratory, Canadensis, PA) or rabbit polyclonal anti-GFPmut2 (diluted 1:1000) as primary antibodies and anti-rabbit IgG antibodies conjugated to Alexa Fluor 488 as a secondary antibody (diluted 1:500).

**Comparative RNA-seq of oocyst and salivary gland sporozoites.** For all oocyst sporozoite and salivary gland sporozoite replicates, RNA was prepared using the Qiagen RNeasy kit with two sequential DNaseI on-column digests, and was quality controlled by analysis on a BioAnalyzer. Barcoded libraries were created using the Illumina TruSeq Stranded mRNA Library Prep Kit, according to the manufacturer's protocol. Sequencing was conducted on an Illumina HiSeq 2500 using 100 nt single read length on three biological replicates per sample type. The resulting data was mapped to the respective reference genomes (*P. yoelii* 17XNL

strain, plasmodb.org v30; *P. falciparum* 3D7 strain, plasmodb.org v30) using Tophat2 in a local Galaxy instance (version 2.1.0). Count files were generated using htseq-count (version 0.6.1) with a minimum alignment quality value set at 30 and a union mode setting. These count files compare the aligned BAM files to a reference GFF file (plasmodb.org v30 for both Py17xNL and Pf3D7) to evaluate the number of reads mapping to each feature, or gene. The count files are combined and compared across conditions using DEseq2 (version 2.11.38), which outputs complete transcript abundance comparisons and performs best among current differential expression tools for three biological replicates. Normalization values for these data are determined by DEseq2 across compared datasets for oocyst sporozoites and salivary gland sporozoites. Statistical metrics utilized were generated by DEseq2[73]. The average number of counts across biological replicates and their standard error of the mean were calculated to allow ranking of transcripts detected over background. Gene ontology terms (components, functions, and processes) were retrieved from PlasmoDB.org (v30). RNA-seq data reported here is available through the GEO depository (Accession #GSE113582).

**MS-based proteomics of *Plasmodium* sporozoites.** Purified oocyst sporozoites were subjected to SDS-PAGE pre-fractionation and in-gel tryptic digestion[31,32]. Briefly, samples were electrophoresed through a 4–20% w/v SDS-polyacrylamide gel (Pierce Precise Tris-HEPES). Gels were stained with Imperial Stain (Thermo Fisher Scientific), de-stained in Milli-Q Water (Millipore), and cut into equal-sized fractions (26 fractions pooled into 13 LC–MS samples for the *P. yoelii* gel and 24 fractions analyzed as 24 LC–MS samples for the *P. falciparum* gel). Gel pieces were then destained with 50 mM ammonium bicarbonate (ABC) in 50% acetonitrile (ACN) and dehydrated with ACN. Disulfide bonds were reduced with 10 mM DTT and cysteines were alkylated with 50 mM iodoacetamide in 100 mM ABC. Gel pieces were washed with ABC in 50% ACN, dehydrated with ACN, and rehydrated with 6.25 ng per µL sequencing grade trypsin (Promega). After incubating overnight at 37 °C, the supernatant was recovered and peptides were extracted by incubating the gel pieces with 2% v/v ACN/1% v/v formic acid, then ACN. The extractions were combined with the digest supernatant, evaporated to dryness in a centrifugal vacuum concentrator, and reconstituted in LC loading buffer consisting of 2% v/v ACN/0.2% v/v trifluoroacetic acid (TFA).

LC was performed using an Agilent 1100 nano pump with electronically controlled split flow at 300 nL per min (*P. falciparum* sample) or an Eksigent nanoLC at 500 nL per min (*P. yoelii* sample)[31,33]. Peptides were separated on a column with an integrated fritted tip (360 µm outer diameter (O.D.), 75 µm inner diameter (I.D.), 15 µm I.D. tip; New Objective) packed in-house with a 20 cm bed of C18 (Dr. Maisch ReproSil-Pur C18-AQ, 120 Å, 3 µm). Prior to each run, sample was loaded onto a trap column consisting of a fritted capillary (360 µm O.D., 150 µm I.D.) packed with a 1 cm bed of the same stationary phase and washed with loading buffer. The trap was then placed in-line with the separation column for the separation gradient. The LC mobile phases consisted of buffer A (0.1% v/v formic acid in water) and buffer B (0.1% v/v formic acid in ACN). The separation gradient was 5% B to 35% B over 60 min (*P. falciparum* sample) or 90 min (*P. yoelii* sample). Tandem MS (MS/MS) was performed with a Thermo Fisher Scientific LTQ Velos Pro-Orbitrap Elite (*P. falciparum*) or LTQ Velos-Orbitrap (*P. yoelii*). Data-dependent acquisition was employed to select the top 20 precursors for collision-induced dissociation and analysis in the ion trap. Dynamic exclusion and precursor charge state selection were employed. Three nanoLC–MS technical replicates were performed for each fraction.

The raw MS data from our previously reported analysis of salivary gland sporozoites[31] were reanalyzed using the same databases and parameters described here. Mass spectrometer output files were converted to mzML format using msConvert version 3.0.6002[74] and searched with Comet version 2015.02 rev.0[75]. The precursor mass tolerance was ±20 ppm, and fragment ions bins were set to a tolerance of 1.0005 *m/z* and a monoisotopic mass offset of 0.4 *m/z*. Semitryptic peptides and up to two missed cleavages were allowed. The search parameters included a static modification of +57.021464 Da at Cys for formation of S-carboxamidomethyl-Cys by iodoacetamide and potential modifications of +15.994915 Da at Met for oxidation, −17.026549 Da at peptide N-terminal Gln for deamidation from formation of pyroGlu, −18.010565 Da at peptide N-terminal Glu for loss of water from formation of pyroGlu, −17.026549 Da at peptide N-terminal Cys for deamidation from formation of cyclized N-terminal S-carboxamidomethyl-Cys, and +42.010565 for acetylation at the N-terminus of the protein, either at N-terminal Met or the N-terminal residue after cleavage of N-terminal Met. The spectra were searched against a database comprising either *P. falciparum* 3D7[76] or *P. yoelii yoelii* 17X[77] (PlasmoDB v.30, www.plasmodb.org[78]) appended with *A. stephensi* Indian AsteI2.3[79] (VectorBase, www.vectorbase.org[80]), and a modified version of the common Repository of Adventitious Proteins (v.2012.01.01, The Global Proteome Machine, www.thegpm.org/cRAP) with the Sigma Universal Standard Proteins removed and the LC calibration standard peptide [Glu-1] fibrinopeptide B appended. Decoy proteins with the residues between tryptic residues randomly shuffled were generated using a tool included in the TPP and interleaved among the real entries. The MS/MS data were analyzed using the TPP[81] version 5.0.0 Typhoon. Peptide spectrum matches (PSMs) were assigned scores in PeptideProphet, peptide-level scores were assigned in iProphet[82], and protein identifications were inferred with ProteinProphet[83]. In the case that multiple proteins were inferred at equal confidence by a set of peptides, the

inference was counted as a single identification and all relevant protein IDs were listed. Only proteins with ProteinProphet probabilities corresponding to a false-discovery rate (FDR) less than 1.0% (as determined from the ProteinProphet mixture models) were reported.

**Protein quantification**. Relative protein abundance within and between samples was estimated using label-free proteomics methods based on spectral counting. Briefly, the spectral counts for a protein were taken as the total number of high-quality PSMs (identified at an iProphet probability corresponding to an FDR less than 1.0%) that identified the protein. Spectral counts were quantified using the StPeter program in the TPP[84]. The distributed spectral counts model was used to divide PSMs from degenerate peptides (peptides whose sequences were found in multiple proteins in the database) among proteins containing that peptide in a weighted fashion[85]. Relative protein abundance within samples was ranked using the normalized spectral abundance factor[55,86]. Relative protein abundance ratios based on spectral counts were normalized and $p$ values were assigned[32]. The raw and fully analyzed data files for these mass spectrometry-based proteomic experiments have been deposited in PRIDE (Accession # PXD009726, PXD009727, PXD009728, and PXD009729).

**Prediction of tryptic peptides**. The CONSeQuence algorithm was used to identify proteins with detectable fully tryptic peptides with no missed cleavages[87]. A threshold of a Rank score ≥0.5 (derived from the combined predictors) was applied, a cutoff that had a sensitivity >70% with a false positive rate <50% when tested on datasets other than the training data as reported by the developers. Application of this algorithm with this threshold to our published *P. falciparum* salivary gland sporozoite proteome only misidentified 2.9% of all proteins as having no detectable peptides[31].

**Statistical analyses**. Statistical tests used in this study were carried out using DEseq2 (RNA-seq), the TPP and the CONSeQuence algorithm (proteomics) as described above for three biological replicates for each sample type. Measurements of statistical significance ($p$ values, $p$-adjusted values) are provided in Supplementary Data. Gene Ontology (GO) analyses were conducted on PlasmoDB (v44), with enriched GO terms identified through embedded Benjamini and Bonferroni statistical analyses.

**Ethics statement**. All animal care adheres to the Association for Assessment and Accreditation of Laboratory Animal Care (AAALAC) guidelines, and all experiments conformed to approved IACUC protocols at Seattle Childrens (formerly Seattle Biomedical Research Institute, Protocol ID #SK-02 to Stefan Kappe) or at Pennsylvania State University (Protocol ID #42678 to Scott Lindner). To this end, all work with vertebrate animals was conducted in strict accordance with the recommendations in the Guide for Care and Use of Laboratory Animals of the National Institutes of Health with approved Office for Laboratory Animal Welfare (OLAW) assurance.

**Reporting summary**. Further information on research design is available in the Nature Research Reporting Summary linked to this article.

## Data availability

Transcriptomic and proteomic data that support the findings of this study have been deposited in the GEO (Accession #GSE113582)] and PRIDE (Accession # PXD009726, PXD009727, PXD009728, and PXD009729) depositories.

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

## Acknowledgements

We appreciate and acknowledge ongoing scientific discussions with Istvan Albert, Aswathy Sebastian, Manuel Llinás and his research group, Kelly Rios for graphic design assistance, as well as advice given on sequencing services provided by the Penn State

Genomics Core Facility (University Park, PA). The authors would like to thank the Johns Hopkins Malaria Research Institute Insectary and Parasitology core facilities, particularly, Christopher Kizito for expert rearing of mosquitoes, and Drs. Abhai Tripathi and Godfree Mlambo for production of the *P. falciparum*-infected mosquitoes. We are grateful to Bloomberg Philanthropies for support of these core facilities. Research reported in this publication was supported by Penn State start-up funds (SEL), the National Institutes of Health National Institute of Allergy and Infectious Disease (http://www.niaid.nih.gov/) under award numbers 1K22AI101039 (S.E.L.), 1R01AI123341 (S.E.L.), K25AI119229 (K.E.S.), R01AI132359 (P.S.), and R01AI134956 (S.H.I.K.), by the National Institutes of Health National Institute of General Medical Sciences (www.nigms.nih.gov) under award number R01GM087221 (R.L.M.), by the National Institutes of Health National Center for Research Resources under award number S10RR027584 (R.L.M.), by the National Science Foundation (www.nsf.gov) under award number 0923536 (R.L.M. and K.E.S.), and by the Provost's Postdoctoral Diversity Fellowship from Johns Hopkins University (M.S.). The content is solely the responsibility of the authors and does not necessarily represent the official views of the National Institutes of Health, the National Science Foundation or the Bill and Melinda Gates Foundation. The funders had no role in study design, data collection and analysis, decision to publish, or preparation of the paper.

## Author contributions

Conducted experiments: S.E.L., K.E.S., M.S., M.P.W., E.N.V., K.J.H. and A.M.M.; Analyzed data: S.E.L., K.E.S., M.P.W., P.S. and S.H.I.K.; Wrote paper: S.E.L., K.E.S., P.S. and S.H.I.K.; Funded the work: S.E.L., K.E.S., P.S., R.L.M. and S.H.I.K.

## Competing interests

The authors declare no competing interests.
