## [Peer Review File · Nature Communications]

Reviewers' comments:

Reviewer #1 (Remarks to the Author):

Nature Communications manuscript NCOMMS-19-17010-T

Linder et al. Extensive transcriptional.....

General.

This paper uses advanced RNAseq and proteomic approaches to study the control of protein expression during the development of the malarial sporozoite (using parasite species infectious either to rodent or to human hosts). The expanded datasets generated from this work have allowed the authors to further develop the established concept that some proteins expressed in the sporozoite are under translation control. The novel contribution of the paper lies in the identification of two parallel 'classes' of translation control that could underpin overlapping but different phases of development, it is this contribution that merits publication.

In the opinion of this reviewer the paper would benefit from deeper consideration of a) the potential correlation between molecular and known organellar development during sporogony (e.g. ribosomal RNA changes and microneme morphogenesis); b) a more structured discussion of the temporal vs. spatial regulation of development within the vector, and c) the unique correlation between gametocyte and sporozoite stages which both require a 'long-lived resting' stage that is able to respond rapidly at any time, to rapid and potentially lethal changes in their environment as they are carried to the alternate host.

The paper is well written, but the figures need significant modification to make them intelligible to those less conversant with the analysis of 'omics' data (e.g. the axes in figs 1-3)

Specific comment (by line No.)

1. Extensive..... is perhaps not the most informative word, the novel factor is that translation al control is not a single mechanism.

48 Not just the basic biology but also the dynamics of transmission (Correlate with gametocyte)

62 Is not the key point that the inoculation of spz. Is unpredictable to the long-lived cell.

121 & 290 Orthogonal – understood, but isn't it more important to suggest independence.

184 Not just comparative but also technically independent.

212 Here would be a good point to recall the 'rhoptry-microneme' transformation between oocyst and sal. gland spz.

319 It would be very good to add data on the location of the mRNA (see Thompson et al. for the gametocyte), is it nuclear or cytoplasmic?

333-334 sentence is repetitive.

382 Uncovered – No, confirmed would be the correct term.

412-415 Isn't the key point the transition from a long-lived cell in comparatively benign environment A to a severely prejudicial environment B without any prior notice?

444 state temp of mosquito maintenance (to clarify the day of sampling)

517 (ABC) If it must be given an abbreviated name use the chemical formula to retain accuracy.

Figures 1-3.

Whilst fully understandable to those expert in 'omics' analyses the axes of the graphs and their presentation need simple descriptions in plain 'English' to be understood by all those interested in the topic e.g. what is $-\log_2(p\text{-adj})$???. For these 3 figures both the images and the legends need to be explained with much greater clarity.

Reviewer #2 (Remarks to the Author):

The paper by Lindner et al provides a comprehensive analysis of the transcriptional and proteomic changes during the maturation of the malaria parasite sporozoite. Taking advantage of the significantly improved technology this paper is able to provide a range of important new insights to aid

our understanding of sporozoite maturation. The work also convincingly shows that there are different translational programs acting during maturation.

Overall, this work is convincing and important. However, there are a number of issues that should still be addressed.

1. There are a number of studies on MAEBL that have indicated that differential expression is important during sporozoite maturation that should be mentioned.
2. Alternative splicing has also been suggested in prior work to play a potential regulatory role. Considering the power of the RNAseq approach used here it would be of great interest to provide an overview of alternative splicing during sporozoite maturation.
3. The authors highlight that the comparative approaches between human and rodent infectious species is of particular value. I fully agree with this and a recent study by Hoo et al., already highlighted the significant variation in transcription observed between species. I am therefore somewhat surprised that the authors did not further explore the biological relevance between proteins whose expression is conserved across the species or divergent.
4. In line 254-260 the authors suggest that environmental differences (different species of mosquito) might drive the need for different translational programs. While this makes a lot of sense – I am wondering why this different program would be used in the same mosquito (*A. stephensi*) species that is used for laboratory based studies, clearly both species can infect and mature in this species of mosquito effectively.
5. It would be good if the authors would show for the proteins investigated further in Figure 4, the expression data for oocyst sporozoites, salivary gland sporozoites and infected hepatocytes. This would provide the readers with a better understanding of relating the omics data with individual genes. It would also make sense to provide a separate panel for these proteins that provides the RNA and protein quantification by RNAseq and MS.
6. The authors provide the quantitative data for the RNAseq and Proteomics in Fig 1 and 2. I think it would be valuable to plot the RNA change vs Protein change as well as this would visualize the findings more clearly

Reviewer #3 (Remarks to the Author):

Review of “Extensive Transcriptional and Translational Regulation Occur During the Maturation of Malaria Parasite Sporozoites”

The study by Linder and Swearingen et al refines our understanding of sporozoite development in the mosquito vector by profiling the transcriptome and proteome of oocyst sporozoites and salivary gland sporozoites to get a comprehensive understanding of transcriptional and translational regulation of these parasite stages that are critical for transmission. These stages are still a relatively understudied component of the Plasmodium life cycle, and offer a huge potential for transmission blocking interventions given their subsequent interactions with the mammalian host at low numbers (bottleneck of the life cycle). The data presented in this paper are a solid contribution to the field, and the paper is written in an engaging and informative way. I have only minor comments regarding the figures and data analysis.

I found figures 1 and 2 a little information light. I think there are more comparisons here that are discussed in the text that could be displayed quantitatively. A few suggestions:

Label outlier genes. For example, probably the top right corner gene in fig 1 a and b is UIS4 as discussed in the text. Having the extreme points labelled will better help the biological interpretation of this figure.

In fig 1, plot the fold change of orthologs against each other and use this plot to label genes that are falciparum or yoelli specific in expression differences. Perhaps this could also be done in fig 2 with the total proteins from each stage (ooSpz, sgSpz).

Plot the relationship between total protein and mRNA for each stage and color by the genes that are in

each translational repression program. It would be helpful to get a visual of where these genes sit in the distribution. My concern is that the functional enrichment of these genes is just an artefact of the fact that these are the most highly expressed genes. It is necessary to see a plot that shows for most transcripts there is a strong relationship between mRNA and protein. The interpretation of the paper is that there is specific repression of these transcripts given their function in mammalian host cell invasion. However, if there is not a clear relationship between transcript level and protein level for other genes, then this is just a random selection of the top expressed genes.

GO analysis (related to point 3 above): It is reported that there is enrichment for GO terms specific to host cell invasion for the genes in each translation repression program. My concern is, same as above, that this could be an artefact of the selection criteria (transcripts must be in the top decile of mRNAs). The GO analysis should have the "gene universe" set to only those transcripts in the top 10% to actually show that there is a specific enrichment for those that are repressed. Perhaps this was done, but more detail on the GO analysis is required in the methods.

Virginia Howick

Response to Referees' Comments:

We thank the reviewers for their very positive and constructive review of our manuscript. We have implemented many of the suggested changes. See detailed responses to each comment below.

Reviewer #1 (Remarks to the Author):

Nature Communications manuscript NCOMMS-19-17010-T

Linder et al. Extensive transcriptional.....

General.

This paper uses advanced RNAseq and proteomic approaches to study the control of protein expression during the development of the malarial sporozoite (using parasite species infectious either to rodent or to human hosts). The expanded datasets generated from this work have allowed the authors to further develop the established concept that some proteins expressed in the sporozoite are under translation control. The novel contribution of the paper lies in the identification of two parallel 'classes' of translation control that could underpin overlapping but different phases of development, it is this contribution that merits publication.

In the opinion of this reviewer the paper would benefit from deeper consideration of a) the potential correlation between molecular and known organellar development during sporogony (e.g. ribosomal RNA changes and microneme morphogenesis); b) a more structured discussion of the temporal vs. spatial regulation of development within the vector, and c) the unique correlation between gametocyte and sporozoite stages which both require a 'long-lived resting' stage that is able to respond rapidly at any time, to rapid and potentially lethal changes in their environment as they are carried to the

alternate host.

The paper is well written, but the figures need significant modification to make them intelligible to those less conversant with the analysis of 'omics' data (e.g. the axes in figs 1-3)

Response: We have used these recommendations to further clarify the figure legends and text so that it is more approachable to all readers. Specifically, plot axes have been more clearly labeled and defined in the figure legends, and key transcripts/proteins in Figures 1 and 2 have been labeled to illustrate the importance of the trends being visualized in the plots.

Specific comment (by line No.)

1. Extensive..... is perhaps not the most informative word, the novel factor is that translational control is not a single mechanism.

Response: In this manuscript we demonstrate several aspects of gene regulation in Plasmodium sporozoites: transcriptional control, translational control, and multi-tiered translational regulation. We feel the current title captures much of this without exceeding the Journal word limit.

48 Not just the basic biology but also the dynamics of transmission (Correlate with gametocyte)

Response: In response to the reviewer's suggestion, this section now reads as follows (new text underlined): "Accomplishing this goal will require greater knowledge of the basic biology and transmission dynamics of the gametocyte as well as pre-erythrocytic sporozoite stages and liver stage parasites. Promising whole-parasite vaccine candidates, based on the sporozoite form of the parasite, are on the horizon and might get closer to realizing a protective vaccine."

62 Is not the key point that the inoculation of spz. Is unpredictable to the long-lived cell.

Response: We have added the following text to capture this concept here: "Within the glands, sporozoites await transmission as long-lived, poised salivary gland sporozoites, which occurs when the mosquito takes its next blood meal and injects these sporozoites into the skin."

121 & 290 Orthogonal – understood, but isn't it more important to suggest independence.

Response: We hold that the term "orthogonal" includes independence as part of its definition, and retain our current text accordingly.

184 Not just comparative but also technically independent.

Response: We have added the following text to capture this concept here: "These findings underscore the strengths and importance of using comparative and independent approaches with human- and rodent-infectious species to identify the important and conserved molecular components of infection."

212 Here would be a good point to recall the 'rhoptry-microneme' transformation between oocyst and sal. gland spz.

Response: We have added the following text to capture this concept here: "This coincides with the

maturation of sporozoite invasion organelles during sporozoite transition (Table 2, Supplemental Table 1, Supplemental Figure 1).”

319 It would be very good to add data on the location of the mRNA (see Thompson et al. for the gametocyte), is it nuclear or cytoplasmic?

Response: We are not clear what is meant here, as the typical life of mRNA includes nuclear and cytosolic phases. Additionally, we were not able to identify the reference that was mentioned here. Because of this, we have not made adjustments.

333-334 sentence is repetitive.

Response: Based on the reviewer’s suggestion, the sentence now reads as follows: “By IFA, nearly all day 14 salivary gland sporozoites showed no UIS4 protein detectable above background (Figure 4A).”

382 Uncovered – No, confirmed would be the correct term.

Response: We respectfully disagree with this point. This is the first report of the extent of translational repression that occurs in sporozoites, that this occurs in both *Plasmodium falciparum* and *Plasmodium yoelii*, and that it occurs in both oocyst sporozoite and salivary gland sporozoite stages. We therefore prefer to maintain the current text.

412-415 Isn’t the key point the transition from a long-lived cell in comparatively benign environment A to a severely prejudicial environment B without any prior notice?

Response: In response to the reviewer’s suggestion, this section now reads as follows (new text underlined): “This regulatory system aligns with the several windows of functionality that are required for sporozoites’ journey as they egress from the relatively benign environment of oocysts on the mosquito midgut, migrate through the hemocoel, invade the salivary glands, remain there poised for transmission and when transmitted, migrate in mammalian tissue, avoid the dangers of the host immune response, traverse cells, and ultimately infect hepatocytes.”

444 state temp of mosquito maintenance (to clarify the day of sampling)

Response: We have added this text to clarify our sporozoite production conditions. Per the reviewer’s suggestion, we have added the following text (underlined): “Wild-type *Plasmodium yoelii* (17XNL strain) sporozoites were produced as previously described in a temperature (24C), humidity (70%) and light (12 hour cycles) controlled incubator.”

517 (ABC) If it must be given an abbreviated name use the chemical formula to retain accuracy.

Response: We respectfully disagree. This abbreviation for ammonium bicarbonate is common in the proteomics literature, and is no less appropriate than the other abbreviations used, e.g., ACN for acetonitrile, DTT for dithiothreitol, and TFA for trifluoroacetic acid.

Figures 1-3.

Whilst fully understandable to those expert in ‘omics’ analyses the axes of the graphs and their

presentation need simple descriptions in plain 'English' to be understood by all those interested in the topic e.g. what is $-\log_2(p\text{-adj})$???. For these 3 figures both the images and the legends need to be explained with much greater clarity.

Response: We have used these recommendations to further clarify the figure legends and text so that it is more approachable to all readers. Plot axes have been more clearly labeled and defined in the figure legends. We have expanded the figure legends to better explain that fold change and statistical significance are plotted to aid comprehension for all readers. Key transcripts/proteins in Figures 1 and 2 have been labeled to illustrate the importance of the trends being visualized in the plots. Moreover, we have also revised Figure 5 (a model figure) to describe these gene regulation programs more fully as well.

Reviewer #2 (Remarks to the Author):

The paper by Lindner et al provides a comprehensive analysis of the transcriptional and proteomic changes during the maturation of the malaria parasite sporozoite. Taking advantage of the significantly improved technology this paper is able to provide a range of important new insights to aid our understanding of sporozoite maturation. The work also convincingly shows that there are different translational programs acting during maturation.

Overall, this work is convincing and important. However, there are a number of issues that should still be addressed.

Response: We appreciate these positive and constructive remarks, and have revised the manuscript to capture the suggested improvements wherever feasible and possible.

1. There are a number of studies on MAEBL that have indicated that differential expression is important during sporozoite maturation that should be mentioned.

Response: We have now specifically described and cited the previous work on MAEBL in sporozoites to tie those stories with the current one. This includes references to it being alternatively spliced and its role in salivary gland invasion.

2. Alternative splicing has also been suggested in prior work to play a potential regulatory role. Considering the power of the RNAseq approach used here it would be of great interest to provide an overview of alternative splicing during sporozoite maturation.

Response: We have explored what is feasible with the current RNA-seq datasets and transcript annotations through consultation with a resident leader in sequencing bioinformatics (Istvan Albert, Penn State). The current state of gene annotations, along with the absence of long read transcriptomic data, would not allow for a robust assessment of alternative splicing in sporozoites. This is a fascinating topic that has recently be addressed in a stand-alone study by Geoff McFadden and Stuart Ralph in Genome Biology demonstrating that it is required for Plasmodium berghei gametocyte maturation. Because of this, we feel that this biological question is outside of the scope of the current publication and

certainly deserving of a stand-alone manuscript when additional, long-read transcriptomic data is available for sporozoites.

3. The authors highlight that the comparative approaches between human and rodent infectious species is of particular value. I fully agree with this and a recent study by Hoo et al., already highlighted the significant variation in transcription observed between species. I am therefore somewhat surprised that the authors did not further explore the biological relevance between proteins whose expression is conserved across the species or divergent.

Response: In this manuscript we have explored a few of the mRNAs and proteins whose expression profiles are conserved across conditions (e.g. species, stages of sporozoite maturation), as those were the best characterized to date. Because a great number of these mRNAs and proteins have divergent expression patterns, we feel that a discussion of specific gene products of this type would not be very informative.

4. In line 254-260 the authors suggest that environmental differences (different species of mosquito) might drive the need for different translational programs. While this makes a lot of sense – I am wondering why this different program would be used in the same mosquito (*A. stephensi*) species that is used for laboratory based studies, clearly both species can infect and mature in this species of mosquito effectively.

Response: Here we intended this to refer to the two different mammalian hosts (human, mouse) and have included text to clarify this in the revised manuscript.

5. It would be good if the authors would show for the proteins investigated further in Figure 4, the expression data for oocyst sporozoites, salivary gland sporozoites and infected hepatocytes. This would provide the readers with a better understanding of relating the omics data with individual genes. It would also make sense to provide a separate panel for these proteins that provides the RNA and protein quantification by RNAseq and MS.

Response: We think this is an excellent idea and have now included the expression profiles for these three gene products compared to TRAP as an unregulated control as Figure 4E.

6. The authors provide the quantitative data for the RNAseq and Proteomics in Fig 1 and 2. I think it would be valuable to plot the RNA change vs Protein change as well as this would visualize the findings more clearly

Response: We have created these plots and now provide them as Supplemental Figure 1.

Reviewer #3 (Remarks to the Author):

Review of “Extensive Transcriptional and Translational Regulation Occur During the Maturation of Malaria Parasite Sporozoites”

The study by Linder and Swearingen et al refines our understanding of sporozite development in the mosquito vector by profiling the transcriptome and proteome of oocyst sporozoites and salivary gland sporozoites to get a comprehensive understanding of transcriptional and translational regulation of these parasite stages that are critical for transmission. These stages are still a relatively understudied component of the Plasmodium life cycle, and offer a huge potential for transmission blocking interventions given their subsequent interactions with the mammalian host at low numbers (bottleneck of the life cycle). The data presented in this paper are a solid contribution to the field, and the paper is written in an engaging and informative way. I have only minor comments regarding the figures and data analysis.

Response: We appreciate these comments about our findings and manuscript, and have added additional figures and descriptions to address the points below.

I found figures 1 and 2 a little information light. I think there are more comparisons here that are discussed in the text that could be displayed quantitatively. A few suggestions: Label outlier genes. For example, probably the top right corner gene in fig 1 a and b is UIS4 as discussed in the text. Having the extreme points labelled will better help the biological interpretation of this figure. In fig 1, plot the fold change of orthologs against each other and use this plot to label genes that are falciparum or yoelli specific in expression differences. Perhaps this could also be done in fig 2 with the total proteins from each stage (ooSpz, sgSpz).

Response: We have labeled notable outliers in Figures 1 and 2 to aid reader comprehension and to place this in context of what is already known in the field. As there many syntenic orthologues between P. falciparum and P. yoelii, most of which do not have similarly regulated gene products, we have opted to retain these comparisons in the supplemental tables instead of generating a new plot.

Plot the relationship between total protein and mRNA for each stage and color by the genes that are in each translational repression program. It would be helpful to get a visual of where these genes sit in the distribution. My concern is that the functional enrichment of these genes is just an artefact of the fact that these are the most highly expressed genes. It is necessary to see a plot that shows for most transcripts there is a strong relationship between mRNA and protein. The interpretation of the paper is that there is specific repression of these transcripts given their function in mammalian host cell invasion. However, if there is not a clear relationship between transcript level and protein level for other genes, then this is just a random selection of the top expressed genes.

Response: We have generated this plot for both species and sample types (4 plots) and have now provided this as Supplemental Figure 2. We have added the following text to the manuscript: "Through comparison of the combined RNA-seq and proteomics datasets, we observed that, as expected, transcript and protein abundance correlated well for many essential and conserved gene products, e.g., CSP, TRAP, CeITOS, SPELD and GEST. However, there was also widespread temporal dysregulation between transcript and protein abundance, including evidence that translational repression is extensively imposed upon many of the most abundant mRNAs of both oocyst sporozoite and salivary gland sporozoite stages of both species (Supplemental Table S6, Supplemental Figure 2)." This plot demonstrates that the abundance of transcripts does not correlate with whether complete, partial, or no translational repression is imposed. This is not surprising, as other studies with Plasmodium and other eukaryotes have demonstrated a similar lack of correlation. Moreover, our inclusion of UIS12 (which is at

the 80th percentile of mRNAs by abundance) in our validation experiments further confirms that this is not a phenomenon restricted only to the top decile of transcripts.

GO analysis (related to point 3 above): It is reported that there is enrichment for GO terms specific to host cell invasion for the genes in each translation repression program. My concern is, same as above, that this could be an artefact of the selection criteria (transcripts must be in the top decile of mRNAs). The GO analysis should have the “gene universe” set to only those transcripts in the top 10% to actually show that there is a specific enrichment for those that are repressed. Perhaps this was done, but more detail on the GO analysis is required in the methods.

Response: We have clarified the Methods related to GO term enrichment, which was conducted using the embedded tool within PlasmoDB (hosted by EuPathDB). This tool provides a list of enriched gene ontology terms over background, including the number of genes with that term in the results versus the background. Moreover, statistical analyses (p-values, Benjamini and Bonferroni tests) are also provided. All GO terms for these defined categories are provided in Supplemental Table 6 to the right of the gene lists that are included in that category (e.g. mRNAs translationally repressed in both stages of P. yoelii).

REVIEWERS' COMMENTS:

Reviewer #2 (Remarks to the Author):

In this revised paper the authors have addressed most of the concerns raised by the reviewers. I am however surprised that the authors did not attempt to address the questions in relation to alternative splicing as well as divergent RNA and protein divergent expression patterns.

1. Alternative splicing patterns have been determined using short read RNAseq data for plasmodium and while I accept that the current dataset may not be able to identify all the different splice variants it should be of high enough quality to evaluate its prevalence.

2. I do disagree with the response by the authors that "Because a great number of these mRNAs and proteins have a divergent expression patterns, we feel that a discussion of specific gene product of type would not be very informative." In my mind I rather see the divergence of expression as potentially highly informative about evolutionary differences. Importantly, considering how important rodent malaria parasite biology is in relation to informing the research community on *P. falciparum* or *P. vivax* biology significant differences in expression of gene products could indicate potential functional divergence.

Reviewer #3 (Remarks to the Author):

I appreciate the authors work in addressing my previous requests. I am happy with how the figures have been updated and the text has been modified.